# Quantifying gender bias towards politicians in cross-lingual language models

**Karolina Stańczak**[1]*, **Sagnik Ray Choudhury**[1], **Tiago Pimentel**[2], **Ryan Cotterell**[3], **Isabelle Augenstein**[1]

**1** Department of Computer Science, University of Copenhagen, Copenhagen, Denmark, **2** Department of Computer Science and Technology, University of Cambridge, Cambridge, United Kingdom, **3** Department of Computer Science, ETH Zürich, Zürich, Switzerland

* ks@di.ku.dk

**Data Availability Statement:** The data we used is public and uploaded the data used to the repository: https://github.com/copenlu/llm-gender-bias-polit.git.

**Funding:** This work is partly funded by Independent Research Fund Denmark under grant

## Abstract

Recent research has demonstrated that large pre-trained language models reflect societal biases expressed in natural language. The present paper introduces a simple method for probing language models to conduct a multilingual study of gender bias towards politicians. We quantify the usage of adjectives and verbs generated by language models surrounding the names of politicians as a function of their gender. To this end, we curate a dataset of 250k politicians worldwide, including their names and gender. Our study is conducted in seven languages across six different language modeling architectures. The results demonstrate that pre-trained language models' stance towards politicians varies strongly across analyzed languages. We find that while some words such as *dead*, and *designated* are associated with both male and female politicians, a few specific words such as *beautiful* and *divorced* are predominantly associated with female politicians. Finally, and contrary to previous findings, our study suggests that larger language models do not tend to be significantly more gender-biased than smaller ones.

## 1 Introduction

In the last decades, digital media has become a primary source of information about political discourse [1] with a dominant share of discussions occurring online [2]. The Internet and social media especially are able to shape public sentiment towards politicians [3], which, in an extreme case, can influence election results [4], and, thus, the composition of a country's government [5]. However, information presented online is subjective, biased, and potentially harmful as it may disseminate misinformation and toxicity. For instance, Prabhakaran et al. [6] show that online comments about politicians, in particular, tend to be more toxic than comments about people in other occupations.

Relatedly, natural language processing (NLP) models are increasingly being used across various domains of the Internet (*e.g.*, in search) [7] and social media (*e.g.*, to translate posts) [8]. These models, however, are typically trained on subjective and imbalanced data. Thus, while they appear to successfully learn general formal properties of the language (*e.g.*, syntax, semantics [9, 10]), they are also susceptible to learning potentially harmful associations [6]. In

agreement number 9130-00092B. The funders had no role in study design, data collection and analysis, decision to publish, or preparation of the manuscript.

**Competing interests:** The authors have declared that no competing interests exist.

particular, pre-trained language models are shown to perpetuate and amplify societal biases found in their training data [11]. For instance, Shwartz et al. [12] showed that pre-trained language models associated negativity with a certain name if the name corresponded to an entity who was frequently mentioned in negative contexts (*e.g.*, Donald for Donald Trump). This strongly suggests a risk of harm when employing language models on downstream tasks such as search or translation.

One such harm that a language model could propagate is that of gender bias [13]. In fact, pre-trained language models have been reported to encode gender bias and stereotypes [11, 14, 15]. Most previous work examining gender bias in language models has focused on English [16], with only a few notable exceptions in recent years [17–20]. The approaches taken in prior work have relied on a range of methods including causal analysis [21], statistical measures such as association tests [14, 22], and correlations [23]. Their findings indicate that gender biases that exist in natural language corpora are also reflected in the text generated by language models.

Gender bias has been examined in stance analysis approaches, but with most investigations focusing on natural language corpora as opposed to language models. For instance, Ahmad et al. [24] and Voigt et al. [25] explicitly controlled for gender bias in two small-scale natural language corpora that focused on politicians within a single country. Specifically, according to Ahmad et al. [24] the media coverage given to male and female candidates in Irish elections did not correspond to the ratio of male to female contestants, with male candidates receiving more coverage. Perhaps surprisingly, Voigt et al. [25] found that there is a smaller difference in the sentiment of responses written to male and female politicians, as opposed to other public figures. However, it is unclear whether these findings would generalize when tested at scale (*i.e.*, examining political figures from around the world) and in text generated by language models.

In this paper, we present a large-scale study on quantifying gender bias in language models with a focus on stance towards politicians. To this end, we generate a dataset for analyzing stance towards politicians encoded in a language model, where stance is inferred from simple grammatical constructs (*e.g.*, "⟨BLANK⟩ PERSON" where ⟨BLANK⟩ is an adjective or a verb). Moreover, we make use of a statistical method to measure gender bias—namely, a latent-variable model—and adapt this to language models. Further, while prior work has focused on monolingual language models [14, 23], we present a fine-grained study of gender bias in six multilingual language models across seven languages, considering 250k politicians from the majority of the world's countries.

In our experiments, we find that, for both male and female politicians, the stance (whether the generated text is written in favor of, against, or neutral) towards politicians in pre-trained language models is highly dependent on the language under consideration. For instance, we show that, while male politicians are associated with more negative sentiment in English, the opposite is true for most other languages analyzed. However, we find no patterns for non-binary politicians (potentially due to data scarcity). Moreover, we find that, on the one hand, words associated with male politicians are also used to describe female politicians; but on the other hand, there are specific words of all sentiments that are predominantly associated with female politicians, such as *divorced*, *maternal*, and *beautiful*. Finally, and perhaps surprisingly, we do not find any significant evidence that larger language models tend to be more gender-biased than smaller ones, contradicting previous studies [14].

## 2 Background

### 2.1 Gender bias in pre-trained language models

Pre-trained language models have been shown to achieve state-of-the-art performance on many downstream NLP tasks [26–30]. During their pre-training, such models can partially

learn a language's syntactic and semantic structure [31, 32]. However, alongside capturing linguistic properties, such as morphology, syntax, and semantics, they also perpetuate and even potentially amplify biases [11]. Consequently, research on understanding and guarding against gender bias in pre-trained language models has garnered an increasing amount of research attention [16], which has created a need for datasets suitable for evaluating the extent to which biases occur in such models. Prior datasets for bias evaluation in language models have mainly focused on English and many revolve around mutating templated sentences' noun phrases, *e. g.*, "This is a(n) ⟨BLACK⟩ PERSON." or "PERSON is ⟨BLACK⟩.", where ⟨BLACK⟩ refers to an attribute such as an adjective or occupation [21–23]. Nadeem et al. [14] and Nangia et al. [15] present an alternative approach to gathering data for analyzing biases in language models. In this approach, crowd workers are tasked with producing variations of sentences that exhibit different levels of stereotypes, *i.e.*, a sentence that stereotypes a particular demographic, a minimally edited sentence that is less stereotyping, produces an anti-stereotype, or has unrelated associations. While the template approach suffers from the artificial context of simply structured sentences [33], the second (*i.e.*, crowdsourced annotations) may convey subjective opinions and is cost-intensive if employed for multiple languages. Moreover, while a fixed structure such as "⟨BLACK⟩." may be appropriate for English, this template can introduce bias for other languages. Spanish, for instance, distinguishes between an ephemeral and a continuous sense of the verb "to be", *i.e.*, *estar*, and *ser*, respectively. As such, a structure such as "PERSON está ⟨BLACK⟩." biases the adjectives studied towards ephemeral characteristics. For example, the sentence "Obama está bueno (Obama is [now] good)" implies that Obama is good-looking as opposed to having the quality of being good. The lexical and syntactic choices in templated sentences may therefore be problematic in a crosslinguistic analysis of bias.

## 2.2 Stance towards politicians

Stance detection is the task of automatically determining if the author of an analyzed text is in favor of, against, or neutral towards a target [34]. Notably, Mohammad et al. [35] observed that a person may demonstrate the same stance towards a target by using negatively or positively sentimented language since stance detection determines the favorability towards a given (pre-chosen) target of interest rather than the mere sentiment of the text. Thus, stance detection is generally considered a more complex task than sentiment classification. Previous work on stance towards politicians investigated biases extant in natural language corpora as opposed to biases in text generated by language models. Moreover, these works mostly targeted specific entities in a single country's political context. Ahmad et al. [24], for instance, analyzed samples of national and regional news by Irish media discussing politicians running in general elections, with the goal of predicting election results. More recently, Voigt et al. [25] collected responses to Facebook posts for 412 members of the U.S. House and Senate from their public Facebook pages, while Pado et al. [36] created a dataset consisting of 959 articles with a total of 1841 claims, where each claim is associated with an entity. In this study, we curated a dataset to examine stance towards politicians worldwide in pre-trained language models.

## 3 Dataset generation

The present study introduces a novel approach to generating a multilingual dataset for identifying gender biases in language models. In our approach, we rely on a simple template "⟨BLACK⟩ PERSON" that allows language models to generate words directly next to entity names. In this case, ⟨BLACK⟩ corresponds to a variable word, *i.e.*, a mask in language modeling terms. This approach imposes no sentence structure and does not suffer from bias introduced by the lexical or syntactical choice of a templated sentence structure (*e.g.*, [21–23]). We argue that this

bottom-up approach can unveil associations encoded in language models between their representations of named entities (NEs) and words describing them. To the best of our knowledge, this method enables the first multilingual analysis of gender bias in language models, which is applicable to any language and with any choice of gendered entity, provided that a list of such entities with their gender is available.

Our approach therefore allows us to examine how the nature of gender bias exhibited in models might differ not only by model size and training data but also by the language under consideration. For instance, in a language such as Spanish, in which adjectives are gendered according to the noun they refer to, grammatical gender might become a highly predictive feature on which the model can rely to make predictions during its pre-training. On the other hand, since inanimate objects are gendered, they might take on adjectives that are not stereotypically associated with their grammatical gender, *e.g.*, "*la espada fuerte* (the strong [feminine] sword)", potentially mitigating the effects of harmful bias in these models.

Given the language independence of our methodology, we conducted analyses on two sets of language models: a monolingual English set and a multilingual set. Overall, our analysis covers seven typologically diverse languages: Arabic, Chinese, English, French, Hindi, Russian, and Spanish. These languages are all included in the training datasets of several well-known multilingual language models (m-BERT [26], XLM [37], and XLM-RoBERTa [38]), and happen to cover a culturally diverse choice of speaker populations.

As shown in Fig 1, our procedure is implemented in three steps. First, we queried the Wikidata knowledge base [39] to obtain politician names in the seven languages under consideration (Section 3.1). Next, using six language models (three monolingual English and three multilingual), we generated adjectives and verbs associated with those politician names (Section 3.2). Finally, we collected sentiment lexica for the analyzed languages to study differences in sentiment for generated words (Section 3.3). We make our dataset publically available for use in future studies (https://github.com/copenlu/llm-gender-bias-polit.git).

## 3.1 Politician names and gender

In the first step of our data generation pipeline (Part 1 in Fig 1), we curated a dataset of politician names and corresponding genders as reported in Wikidata entries for political figures. We restricted ourselves to politicians with a reported date of birth before 2001 and who had information regarding their gender on Wikidata. We note that politicians whose gender information was unavailable account for <3% of the entities for all languages. We also note that not

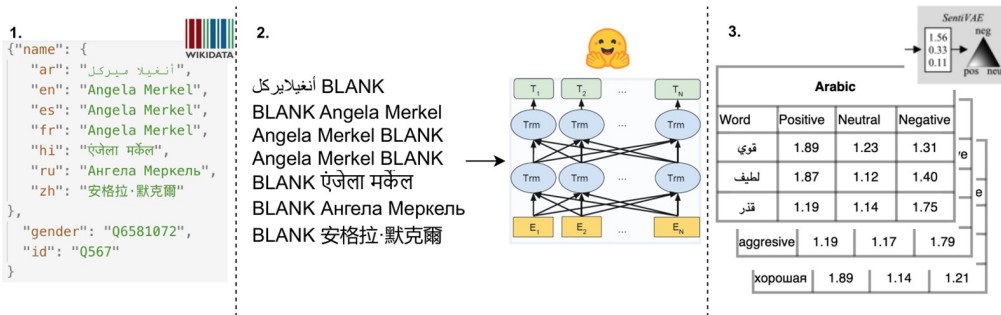

**Fig 1. The three-part dataset generation procedure.** Part 1 depicts politician names and their gender in the seven analyzed languages. Part 2 depicts the adjectives and verbs associated with the names that are generated by the language model. Part 3 depicts the sentiment lexica with associated values for each word.

**Table 1. Counts of politicians grouped by their gender according to Wikidata (female, male, non-binary) for each language.**

| Gender | Languages | | | | | | |
|---|---|---|---|---|---|---|---|
| | Arabic | Chinese | English | French | Hindi | Russian | Spanish |
| male | 206.526 | 207.713 | 206.493 | 233.598 | 206.778 | 208.982 | 226.492 |
| female | 44.962 | 45.683 | 44.703 | 53.437 | 44.958 | 45.277 | 50.888 |
| non-binary | 67 | 67 | 66 | 67 | 67 | 67 | 67 |

all names were available on Wikidata in all languages, causing deviations in the counts for different languages (with a largely consistent set of non-binary politicians). Wikidata distinguishes between 12 gender identities: cisgender female, female, female organism, non-binary, genderfluid, genderqueer, male, male organism, third gender, transfeminine, transgender female, and transgender male. This information is maintained by the community and regularly updated. We discuss this further in Section 6. We decided to exclude female and male organisms from our dataset, as they refer to animals that (for one reason or another) were running for elections. Further, we replaced the cisgender female label with the female label. Finally, we created a non-binary gender category, which includes all politicians not identified as male or female (due to the small number of politicians for each of these genders; see S1 Text. Politician Gender in the Appendix). Table 1 presents the counts of politicians grouped by their gender (female, male, non-binary) for each language. (See S1 Text. Politician Gender in the Appendix for the detailed counts across all gender categories.) On average, the male-to-female gender ratio is 4:1 across the languages and there are very few names for the non-binary gender category.

## 3.2 Language generation

In the second step of the data generation process (Part 2 in Fig 1), we employed language models to generate adjectives and verbs associated with the politician's name. Metaphorically, this language generation process can be thought of as a word association questionnaire. We provide the language model with a politician's name and prompt it to generate a token (verb or adjective) with the strongest association to the name. We could take several approaches towards that goal. One possibility is to analyze a sentence generated by the language model which contains the name in question. However, the bidirectional language models under consideration are not trained with language generation in mind and hence do not explicitly define a distribution over language [40]—their pre-training consists of predicting masked tokens in already existing sentences [10]. Goyal et al. [41] proposed generation using a sampler based on the Metropolis-Hastings algorithm [42] to draw samples from non-probabilistic masked language models. However, the sentence length has to be provided in advance, and generated sentences often lack diversity, particularly when the process is constrained by specifying the names. Another possible approach would be to follow Amini et al. [33] and compute the average treatment effect of a politician's name on the adjective (verb) choice given a dependency parsed sentence. In particular, Amini et al. [33] derived counterfactuals from the dependency structure of the sentence and then intervened on a specific linguistic property of interest, such as the gender of a noun. This method, while effective, becomes computationally prohibitive when handling a large number of entities.

In this work, we simplify this problem. We query each language model by providing it with either a "⟨BLACK⟩ PERSON" input or its inverse "PERSON ⟨BLACK⟩," depending on the grammar formalisms of the language under consideration. (See S1 Table. Word Orderings in the Appendix for the word orderings used for each language.) Our approach returns a ranked list of words

(with their probabilities) that the model associates with the name. The ranked list of words included a wide variety of part of speech (POS) categories; however, not all POS categories necessarily lend themselves to analyzing sentiment with respect to an associated name. We therefore filtered the data to just the adjectives and verbs, as these have been shown to capture sentiment about a name [43]. To filter these words, we used the Universal Dependency [44] treebanks, and only kept adjectives and verbs that were present in any of the language-specific treebanks. We then lemmatize the data to prevent us from recovering this trivial gender relationship between the politician's name and the gendered form of the associated adjective or verb.

A final issue is that all our models use subword tokenizers and, therefore, a politician's name is often not just tokenized by whitespace. For example, the name "Narendra Modi" is tokenized as ["na", '##ren', "##dra", "mod", "##i"] by the WordPiece tokenizer [45] in BERT [26]. This presents a challenge in ascertaining whether a name was present in the model's training data from its vocabulary. However, all politicians whose names were processed have a Wikipedia page in at least one of the analyzed languages. As Wikipedia is a subset of the data on which these models were trained (except BERTweet, which is trained on a large collection of 855M English tweets), we assume that the named entities occurred in the language models' training data, and therefore, that the predicted words for the ⟨BLACK⟩ token provide insight into the values reflected by these models.

In total, we queried six language models for the word association task across two setups: a monolingual and a multilingual setup. In the monolingual setup, we used the following English language models: BERT [base and large; 26], BERTweet [46], RoBERTa [base and large; 27], ALBERT [base, large, xlarge and xxlarge; 47], and XLNet [base and large; 28]. In the multilingual setup, we used the following multilingual language models: m-BERT [26], XLM [base and large; 37] and XLM-RoBERTa [base and large; 38]. The pre-training of these models included data for each of the seven languages under consideration. Each language model, together with its corresponding features is listed in Table 2. For each language, we entered the politicians' names as written in that particular language.

**Table 2. Overview of analyzed the language models.**

| Language Model | # of Parameters | Training Data |
|---|---:|---|
| *Monolingual* | | |
| ALBERT-base | 0.11E+08 | Wikipedia, BookCorpus |
| ALBERT-large | 0.17E+08 | Wikipedia, BookCorpus |
| ALBERT-xlarge | 0.58E+08 | Wikipedia, BookCorpus |
| ALBERT-xxlarge | 2.23E+08 | Wikipedia, BookCorpus |
| BERT-base | 1.1E+08 | Wikipedia, BookCorpus |
| BERT-large | 3.4E+08 | Wikipedia, BookCorpus |
| BERTweet | 1.1E+08 | Tweets |
| RoBERTa-base | 1.25E+08 | Wikipedia, BookCorpus, CC-News, OpenWebText, Stories |
| RoBERTa-large | 3.55E+08 | Wikipedia, BookCorpus, CC-News, OpenWebText, Stories |
| XLNet-base | 1.1E+08 | Wikipedia, BookCorpus, Giga5, ClueWeb, CommonCrawl |
| XLNet-large | 3.4E+08 | Wikipedia, BookCorpus, Giga5, ClueWeb, CommonCrawl |
| *Multilingual* | | |
| BERT | 1.1E+08 | Wikipedia |
| XLM-base | 2.5E+08 | Wikipedia |
| XLM-large | 5.7E+08 | Wikipedia |
| XLM-RoBERTa-base | 1.25E+08 | CommonCrawl |
| XLM-RoBERTa-large | 3.55E+08 | CommonCrawl |

### 3.3 Sentiment data

Previously, it has been shown that words used to describe entities differ based on the target's gender and that these discrepancies can be used as a proxy to quantify gender bias [43, 48]. In light of this, we categorized words generated by the language model into positive, negative, and neutral sentiments (Part 3 in Fig 1).

To accomplish this task, we required a lexicon specific to each analyzed language. For English, we used the existing sentiment lexicon of Hoyle et al. [49]. This lexicon is a combination of multiple smaller lexica that has been shown to outperform the individual lexica, as well as their straightforward combination when applied to a text classification task involving sentiment analysis. However, such a comprehensive lexicon was only available for English, we therefore collected various publicly available sentiment lexica for the remaining languages, which we combined into one comprehensive lexicon per language using SentiVAE [49]—a variational autoencoder model (VAE; [50]). VAE allows for unifying labels from multiple lexica with disparate scales (binary, categorical, or continuous). In SentiVAE, the sentiment values for each word from different lexica are 'encoded' into three-dimensional vectors whose sum is added to form the parameters of a Dirichlet distribution over the latent representation of the word's polarity value. From this procedure, we obtained the final lexicon for each language—a list of words present in at least one of the individual lexica and three-dimensional representations of the words' sentiments (positive, negative, and neutral). Through this approach, we aimed to cover more words and create a more robust sentiment lexicon while retaining scale coherence.

Following the results presented in [49], we hypothesized that combining a larger number of individual lexica with SentiVAE leads to more reliable results. We confirmed this assumption for all languages but Hindi. We combined three multilingual sentiment lexica for all remaining languages: the sentiment lexicon by Chen and Skiena [51], BabelSenticNet [52] and UniSent [53]. Due to the poor evaluation performance, we decided to exclude BabelSenticNet and UniSent lexica for Hindi. Instead, we combined the sentiment lexica curated by Chen and Skiena [51], Desai [54], and Sharan [55]. Additionally, we incorporated monolingual sentiment lexica for Arabic [56], Chinese [57, 58], French [59, 60], Russian [61] and Spanish [62–64].

Following Hoyle et al. [49], we evaluated the lexica resulting from the VAE approach on a sentiment classification task by inspecting their performance—for each language. Namely, we used the resulting lexica to automatically label utterances (sentences and paragraphs) for their sentiment, based on the average sentiment of words in each sentence. This is shown in the Appendix in S2 Table. Sentiment Analysis Evaluation where we also include the best performance achieved by a supervised model (as reported in the original dataset's paper) as a point of reference. In general, the sentiment lexicon approach achieves comparable performance to the respective supervised model for most of the analyzed languages. We observed the greatest drop in performance for French, but a performance decrease was also visible for Hindi and Chinese. However, the results in S2 Table. Sentiment Analysis Evaluation in the Appendix are based on the sentiment classification of utterances rather than single words, as in our setup. Here, we treat these results as a lower-bound performance in our single-word scenario.

## 4 Method

Our aim is to quantify the usage of words around the names of politicians as a function of their gender. Formally, let $\mathcal{G} = \{male, female, non-binary\}$ be the set of genders, as discussed in Section 3.1; we denote elements of $\mathcal{G}$ as $g$. Further, let $\mathcal{N}$ be the set of politicians' names found in our dataset; we denote elements of $\mathcal{N}$ as $n$. With $\boldsymbol{w}$ we denote a lemmatized word in a language-specific vocabulary $\boldsymbol{w} \in \mathcal{W}$. Finally, let G, **W** and N be, respectively, gender-, word-

and name-valued random variables, which are jointly distributed according to a probability distribution $p(\mathbf{W} = \boldsymbol{w}, \mathrm{G} = g, \mathrm{N} = n)$. We shall write $p(\boldsymbol{w}, g, n)$, omitting random variables, when clear from the context. Assuming we know the true distribution $p$, there is a straightforward metric for how much the word $\boldsymbol{w}$ is associated with the gender $g$—the point-wise mutual information (PMI) between $\boldsymbol{w}$ and $g$:

$$\mathrm{PMI}(\boldsymbol{w}, g) = \log \frac{p(\boldsymbol{w}, g)}{p(\boldsymbol{w})p(g)} = \log \frac{p(\boldsymbol{w} \mid g)}{p(\boldsymbol{w})} \tag{1}$$

Much like mutual information (MI), PMI quantifies the amount of information we can learn about a specific variable from another, but, in contrast to MI, it is restricted to a single gender–word pair. In particular, as evinced in Eq (1), PMI measures the (log) probability of co-occurrence scaled by the product of the marginal occurrences. If a word is more often associated with a particular gender, its PMI will be positive. For example, we would expect a high value for PMI(*female*, *pregnant*) because the joint probability of these two words is higher than the marginal probabilities of *female* and *pregnant* multiplied together. Accordingly, in an ideal unbiased world, we would expect words such as *successful* or *intelligent* to have a PMI of approximately zero with all genders.

Above, we consider the true distribution $p$ to be known, while, in fact, we solely observe samples from $p$. In the following, we assume that we only have access to an empirical distribution $\tilde{p}$ derived from samples from the true distribution $p$

$$\tilde{p}(\boldsymbol{w}, g, n) \overset{\mathrm{def}}{=} \frac{1}{I} \sum_{i=1}^{I} \mathbb{1}\{\boldsymbol{w} = \boldsymbol{w}_i, g = g_i, n = n_i\} \tag{2}$$

where we assume a dataset $\mathcal{D} = \{\langle \boldsymbol{w}_i, g_i, n_i \rangle\}_{i=1}^{I}$ is composed of $I$ independent samples from the distribution $p$. With a simple plug-in estimator, we can then estimate the PMI above using this $\tilde{p}$, as opposed to $p$. The plug-in estimator, however, may produce biased PMI estimates; these biases are in general positive, as shown by [65, 66].

To get a better approximation of $p$, we estimate a model $p_{\boldsymbol{\theta}}$ to generalize from the observed samples $\tilde{p}$ with the hope that we will be able to better infer the relationship between G and **W**. We estimate $p_{\boldsymbol{\theta}}$ by minimizing the cross-entropy given below

$$\mathcal{L}(\boldsymbol{\theta}) = -\sum_{n \in \mathcal{N}} \sum_{\boldsymbol{w} \in W} \tilde{p}(\mathbf{W} = \boldsymbol{w}, \mathrm{N} = n) \log p_{\boldsymbol{\theta}}(\mathbf{W} = \boldsymbol{w}, \mathrm{G} = g_n) \tag{3}$$

where $g_n$ is the gender of the politician with name $n$. Then, we consider a regularized estimator of pointwise mutual information. We factorize $p_{\boldsymbol{\theta}}(\boldsymbol{w}, g) \overset{\mathrm{def}}{=} p_{\boldsymbol{\eta}}(\boldsymbol{w} \mid g)p_{\phi}(g)$. We first define

$$p_{\boldsymbol{\eta}}(\boldsymbol{w} \mid g) \propto \exp\left(m_{\boldsymbol{w}} + \boldsymbol{f}_g^{\top} \boldsymbol{\eta}_{\boldsymbol{w}}\right) \tag{4}$$

where $\boldsymbol{f}_g \in \{0, 1\}^{|\mathcal{G}|}$ is a one-hot gender representation, and both $\boldsymbol{m} \in \mathbb{R}^{|\mathcal{W}|}$ and $\boldsymbol{\eta} \in \mathbb{R}^{|\mathcal{W}| \times |\mathcal{G}|}$ are model parameters, which we index as $m_{\boldsymbol{w}} \in \mathbb{R}$ and $\boldsymbol{\eta}_{\boldsymbol{w}} \in \mathbb{R}^{|\mathcal{G}|}$; these parameters induce a prior distribution over words $p_{\boldsymbol{\theta}}(\boldsymbol{w}) \propto \exp(m_{\boldsymbol{w}})$ and word-specific deviations, respectively. Second, we define

$$p_{\phi}(g) \propto \exp\left(\phi_g\right) \tag{5}$$

where $\boldsymbol{\phi} \in \mathbb{R}^{|\mathcal{G}|}$ are model parameters, which we index as $\phi_g \in \mathbb{R}$.

Assuming that $p_{\boldsymbol{\eta}}(\boldsymbol{w} \mid g) \approx p(\boldsymbol{w} \mid g)$, *i.e.*, that our model learns the true distribution $p$, we have that $\boldsymbol{f}_g^{\top} \boldsymbol{\eta}_{\boldsymbol{w}}$ will be equivalent (up to an additive term that is constant on the word) to the

PMI in Eq (1):

$$\mathrm{PMI}(\boldsymbol{w}, g) = \log \frac{p(\boldsymbol{w} \mid g)}{p(\boldsymbol{w})} \approx \log \frac{p_{\eta}(\boldsymbol{w} \mid g)}{p_{\theta}(\boldsymbol{w})} \tag{6}$$

$$= \log \frac{\frac{\exp(m_{\boldsymbol{w}} + \boldsymbol{f}_g^{\top} \boldsymbol{\eta}_{\boldsymbol{w}})}{\sum_{\boldsymbol{w}' \in \mathcal{W}} \exp(m_{\boldsymbol{w}'} + \boldsymbol{f}_g^{\top} \boldsymbol{\eta}_{\boldsymbol{w}'})}}{\exp(m_{\boldsymbol{w}})} \tag{7}$$

$$= \log \frac{\exp(\boldsymbol{f}_g^{\top} \boldsymbol{\eta}_{\boldsymbol{w}})}{\sum_{\boldsymbol{w}' \in \mathcal{W}} \exp\left(m_{\boldsymbol{w}'} + \boldsymbol{f}_g^{\top} \boldsymbol{\eta}_{\boldsymbol{w}'}\right)} \tag{8}$$

$$= \boldsymbol{f}_g^{\top} \boldsymbol{\eta}_{\boldsymbol{w}} - \log \sum_{\boldsymbol{w}' \in \mathcal{W}} \exp\left(m_{\boldsymbol{w}'} + \boldsymbol{f}_g^{\top} \boldsymbol{\eta}_{\boldsymbol{w}'}\right) \tag{9}$$

If we estimate the model without any regularization or latent sentiment, then ranking the words by their deviation scores from the prior distribution is equivalent to ranking them by their PMI. However, we are not merely interested in quantifying the usage of words around the entities but are also interested in analyzing those words' sentiments. Thus, let $\mathcal{S} = \{pos, neg, neu\}$ be a set of sentiments; we denote elements of $\mathcal{S}$ as $s$. More formally, the extended model jointly represents adjective (or verb) choice ($\boldsymbol{w}$) with its sentiment ($s$), given a politician's gender ($g$) as follows:

$$p_{\theta}(\boldsymbol{w}, g, s) \stackrel{\text{def}}{=} p_{\eta}(\boldsymbol{w} \mid s, g)\, p_{\sigma}(s \mid g)\, p_{\phi}(g) \tag{10}$$

We compute the first factor in Eq (10) by plugging in Eq (4), albeit with a small modification to condition it on the latent sentiment:

$$p_{\eta}(\boldsymbol{w} \mid s, g) \propto \exp(m_{\boldsymbol{w}} + \boldsymbol{f}_g^{\top} \boldsymbol{\eta}_{\boldsymbol{w}, s}) \tag{11}$$

The second factor in Eq (10) is defined as $p_{\sigma}(s \mid g) \propto \exp(\sigma_{s,g})$, and the third factor is defined as before, *i.e.*, $p_{\phi}(g) \propto \exp(\phi_g)$, where $\sigma_{s,g}, \phi_g \in \mathbb{R}$ are learned. Thus, the model $p_{\theta}$ is parametrized by $\boldsymbol{\theta} = \{\boldsymbol{\eta} \in \mathbb{R}^{|\mathcal{W}| \times |\mathcal{S}| \times |\mathcal{G}|}, \boldsymbol{\sigma} \in \mathbb{R}^{|\mathcal{S}| \times |\mathcal{G}|}, \boldsymbol{\phi} \in \mathbb{R}^{|\mathcal{G}|}\}$, with $\boldsymbol{\eta}_{\boldsymbol{w},s} \in \mathbb{R}^{|\mathcal{G}|}$ denoting the word- and sentiment-specific deviation. As we do not have access to explicit sentiment information (it is encoded as a latent variable), we marginalize it in Eq (10) to construct a latent-variable model

$$p_{\theta}(\boldsymbol{w}, g) = \sum_{s \in \mathcal{S}} p_{\eta}(\boldsymbol{w} \mid s, g)\, p_{\sigma}(s \mid g)\, p_{\phi}(g) \tag{12}$$

whose marginal likelihood we maximize to find good parameters $\boldsymbol{\theta}$. This model enables us to analyze how the choice of a generated word depends not only on a politician's gender but also on a sentiment via jointly modeling gender, sentiment, and generated words as depicted in Fig 2. Through the distribution $p_{\eta}(\boldsymbol{w} \mid s, g)$, this model enables us to extract ranked lists of adjectives (or verbs), grouped by gender and sentiment, that were generated by a language model to describe politicians.

We additionally apply posterior regularization [67] to guarantee that our latent variable corresponds to sentiments. This regularization is taken as the Kullback–Leibler (KL) divergence between our estimate of $p_{\theta}(s \mid \boldsymbol{w})$ and $q(s \mid \boldsymbol{w})$; where $q$ is a target posterior that we obtain from the sentiment lexicon described in detail in Section 3.3. Further, we also use $L_1$-regularization to account for sparsity. Combing the cross-entropy term, with the KL and $L_1$

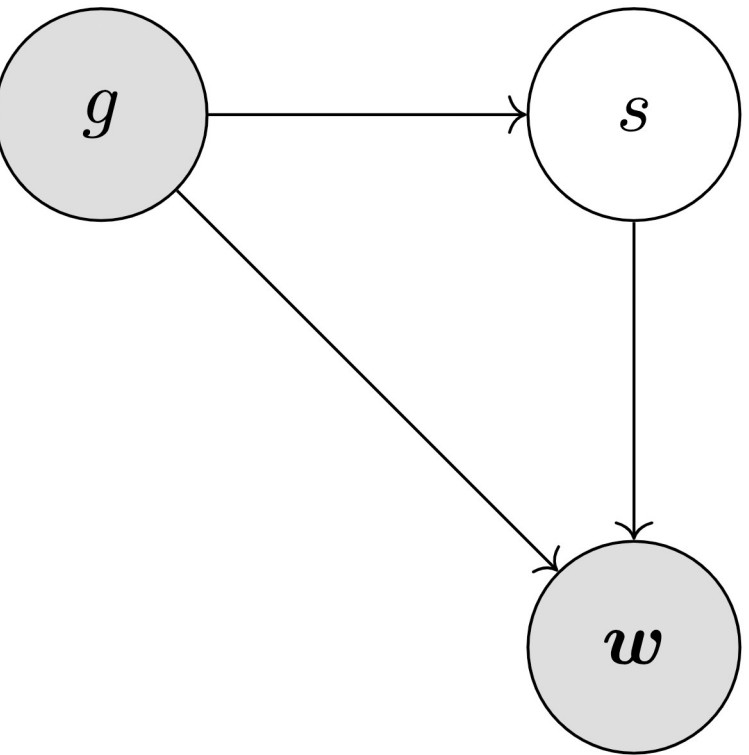

**Fig 2. Graphical model depicting the relations among politician's gender (g), generated word's sentiment (s), and the generated word (w).**

regularizers, we arrive at the loss function:

$$\mathcal{O}(\boldsymbol{\theta}) = \mathcal{L}(\boldsymbol{\theta}) + \alpha \cdot \underbrace{\sum_{w \in W} \sum_{s \in \mathcal{S}} q(s \mid w) \log \frac{q(s \mid w)}{p_{\boldsymbol{\theta}}(s \mid w)}}_{\text{posterior regularizer}} + \beta \cdot \underbrace{(\|\boldsymbol{\eta}\|_1 + \|\boldsymbol{\sigma}\|_1 + \|\boldsymbol{\phi}\|_1)}_{L_1 \text{ regularizer}} \tag{13}$$

with hyperparameters $\alpha, \beta \in \mathbb{R}_{\geq 0}$. This objective $\mathcal{O}$ is minimized with the Adam optimizer [68]. We then validate the method through an inspection of the posterior regularizer values; values close to zero indicate the validity of the approach as a low KL divergence implies our latent distribution $p_{\boldsymbol{\theta}}$ closely represents the lexicon's sentiment.

Finally, we note that due to the relatively small number of politicians identified in the non-binary gender group, we restrict ourselves to two binary genders in the generative latent-variable setting of the extended model. In Section 6, we discuss the limitations of this modeling decision.

## 5 Experiments and results

We applied the methods defined in Section 4 to study the presence of gender bias in the dataset described in Section 3. We hypothesized that the generated vocabulary for English would be much more versatile than for the other languages. Therefore, in order to decrease computational costs and maintain similar vocabulary sizes across languages, we decided to further limit the number of generated words for English. We used the 20 highest probability adjectives and

verbs generated for each politician's name in English, both in mono- and multilingual setups. For the other languages, the top 100 adjectives and top 20 verbs were used. Detailed counts of generated adjectives are presented in S3 Table. Generated Words in the Appendix. We confirmed our hypothesis that the vocabulary generated for English is broader, as including the top 20 adjectives and verbs for English results in a vocabulary set (unique lemmata generated by each of the language models) similar to or bigger than for Spanish—the largest vocabulary of all the remaining languages.

First, using the English portion of the dataset, we analyzed estimated PMI values to look for the words whose association with a specific gender differs the most across the three gender categories. Then, we followed a virtually identical experimental setup as presented in [43] for our dataset. In particular, we tested whether adjectives and verbs generated by language models unveil the same patterns as discovered in natural corpora and if they confirm previous findings about the stance towards politicians. To this end, we employed PMI and the latent-variable model on our data set and qualitatively evaluated the results. We analyzed generated adjectives and verbs in terms of their alignment within supersenses—a set of pre-defined semantic word categories.

Next, we conducted a multilingual analysis for the seven selected languages via PMI and the latent-variable model to inspect both qualitative and quantitative differences in words generated by six cross-lingual language models. Further, we performed a cluster analysis of the generated words based on their word representations extracted from the last hidden state of each language model for all analyzed languages. In additional experiments in Appendix S3 Text. Additional Experiments, we examined gender bias towards the most popular politicians. Then, for each language, we studied gender bias towards politicians whose country of origin (*i. e.*, their nationality) uses the respective language as an official language. Finally, we investigated gender bias towards politicians born before and after the Baby Boom to control for temporal changes. However, we did not find any significant patterns.

Following Hoyle et al. [43], our reported results were an average over hyperparameters: for the $L_1$ penalty $\alpha \in \{0, 10^{-5}, 10^{-4}, 0.001, 0.01\}$ and for the posterior regularization $\beta \in \{10^{-5}, 10^{-4}, 0.001, 0.01, 0.1, 1, 10, 100\}$.

## 5.1 Monolingual setup

**5.1.1 PMI and latent-variable model.**   In the following, we report the PMI values calculated based on words generated by all the monolingual English language models under consideration. From the PMI values for words associated with politicians of male, female, or non-binary genders, it is apparent that words associated with the female gender are often connected to weaknesses such as *hysterical* and *fragile* or to their appearance (*blonde*), while adjectives generated for male politicians tend to describe their political beliefs (*fascist* and *bolshevik*). There is no such distinguishable pattern for the non-binary gender, most likely due to an insufficient amount of data. See Table 3 for details.

The results for the latent-variable model are similar to those for the PMI analysis. Adjectives associated with appearance are more often generated for female politicians. Additionally, words describing marital status (*divorced* and *unmarried*) are more often generated for female politicians. On the other hand, positive adjectives that describe men often relate to their character values such as *bold* and *independent*. Further examples are available in Table 4.

Following Hoyle et al. [43], we used two existing semantic resources based on the WordNet database [69] to quantify the patterns revealed above. We grouped adjectives into 13 supersense classes using classes defined by Tsvetkov et al. [70]; similarly, we grouped verbs into 15

**Table 3. Top 15 adjectives with the biggest difference in PMI for male and female (left); top 5 adjectives (top right) and bottom 5 verbs (bottom right) PMI for non-binary gender.** Based on words generated by all monolingual language models for English.

| word | $\text{PMI}_f$ | $\text{PMI}_m$ | |
|---|---|---|---|
| blonde | 1.7 | -2.2 | |
| fragile | 1.7 | -2.0 | |
| dreadful | 1.6 | -1.3 | |
| feminine | 1.6 | -1.7 | |
| stormy | 1.6 | -1.8 | |
| ambiguous | 1.5 | -1.4 | |
| beautiful | 1.5 | -1.4 | |
| divorced | 1.5 | -2.4 | |
| irrelevant | 1.5 | -1.5 | |
| lovely | 1.5 | -1.6 | |
| marital | 1.4 | -1.8 | |
| pregnant | 1.4 | -2.5 | |
| translucent | 1.4 | -2.1 | |
| bolshevik | -3.1 | 0.1 | |
| capitalist | -3.4 | 0.2 | |
| word | $\text{PMI}_{nb}$ | $\text{PMI}_f$ | $\text{PMI}_m$ |
| smaller | 5.8 | 0.1 | 0.0 |
| militant | 5.7 | -0.3 | 0.0 |
| distinctive | 5.6 | 0.4 | -0.2 |
| ambiguous | 5.2 | 1.5 | -1.4 |
| evident | 5.2 | 0.4 | -0.2 |
| word | $\text{PMI}_{nb}$ | $\text{PMI}_f$ | $\text{PMI}_m$ |
| approach | 5.8 | 0.1 | -0.1 |
| await | 5.3 | 0.5 | -0.2 |
| escape | 5.1 | 0.2 | -0.1 |
| crush | 4.9 | 0.3 | -0.1 |
| capture | 4.8 | -0.3 | 0.0 |

**Table 4. The top 10 adjectives, for female and male politicians, that have the largest average deviation for each sentiment, extracted from all monolingual English models.**

| female | | | | | | male | | | | | |
|---|---|---|---|---|---|---|---|---|---|---|---|
| negative | | neuter | | positive | | negative | | neuter | | positive | |
| divorced | 3.4 | bella | 3.1 | beautiful | 3.2 | stolen | 1.5 | based | 1.2 | bold | 1.4 |
| bella | 3.2 | women | 3.0 | lovely | 3.1 | american | 1.4 | archeological | 1.2 | vital | 1.4 |
| fragile | 3.2 | misty | 3.0 | beloved | 3.1 | forbidden | 1.4 | hilly | 1.1 | renowned | 1.4 |
| women | 3.1 | maternal | 3.0 | sweet | 3.1 | first | 1.4 | variable | 1.1 | mighty | 1.4 |
| couple | 3.0 | pregnant | 3.0 | pregnant | 3.1 | undergraduate | 1.4 | embroider | 1.1 | modest | 1.4 |
| mere | 2.9 | agriculture | 3.0 | female | 3.0 | fascist | 1.4 | filipino | 1.1 | independent | 1.4 |
| next | 2.9 | divorced | 2.9 | translucent | 3.0 | tragic | 1.4 | distinguishing | 1.1 | monumental | 1.3 |
| another | 2.9 | couple | 2.9 | dear | 3.0 | great | 1.4 | retail | 1.1 | like | 1.3 |
| lower | 2.9 | female | 2.9 | marry | 3.0 | insulting | 1.4 | socially | 1.0 | support | 1.3 |
| naughty | 2.9 | blonde | 2.9 | educated | 2.9 | out | 1.4 | bottled | 1.0 | notable | 1.3 |

supersenses according to the database presented in Miller et al. [71]. We list the defined groups together with their respective example words in the Appendix S2 Text. Supersenses.

We performed an unpaired permutation test [72] considering the 100 largest-deviation words and found that male politicians are more often described negatively when using adjectives related to their emotions (*e.g.*, *angry*) while more positively with adjectives related to their minds (*e.g.*, *intelligent*), as presented in Fig 3. These results differ from the findings of Hoyle et al. [43], where no significant evidence of these tendencies was found.

**5.1.2 Sentiment analysis.** We report the results in Fig 4. We found that words more commonly generated by language models that describe male rather than female politicians are also more often negative and that this pattern holds across most language models. However, based on the results of the qualitative study (see details in Table 4), we assume it is due to several strongly positive words such as *beloved* and *marry*, which are highly associated with female politicians. We note that the deviation scores for words associated with male politicians are relatively low compared to the scores for adjectives and verbs associated with female politicians which introduces also more neutral words to the list of words of negative sentiment. Ultimately, this suggests that words of negative and neutral sentiment are more equally distributed across genders with few words being used particularly often in association with a specific gender. Conversely, positive words generated around male and female genders differ more substantially.

To investigate whether there were significant differences across language models based on their size and architecture, we performed a two-way analysis of variance (ANOVA). Language, model size, and architecture were the independent variables and sentiment values were the target variables. We then analyzed the differences in the mean frequency with which the 100 largest deviation words (adjectives and verbs) correspond to each sentiment for the male and female genders. The results presented in Table 5 indicate significant differences in negative

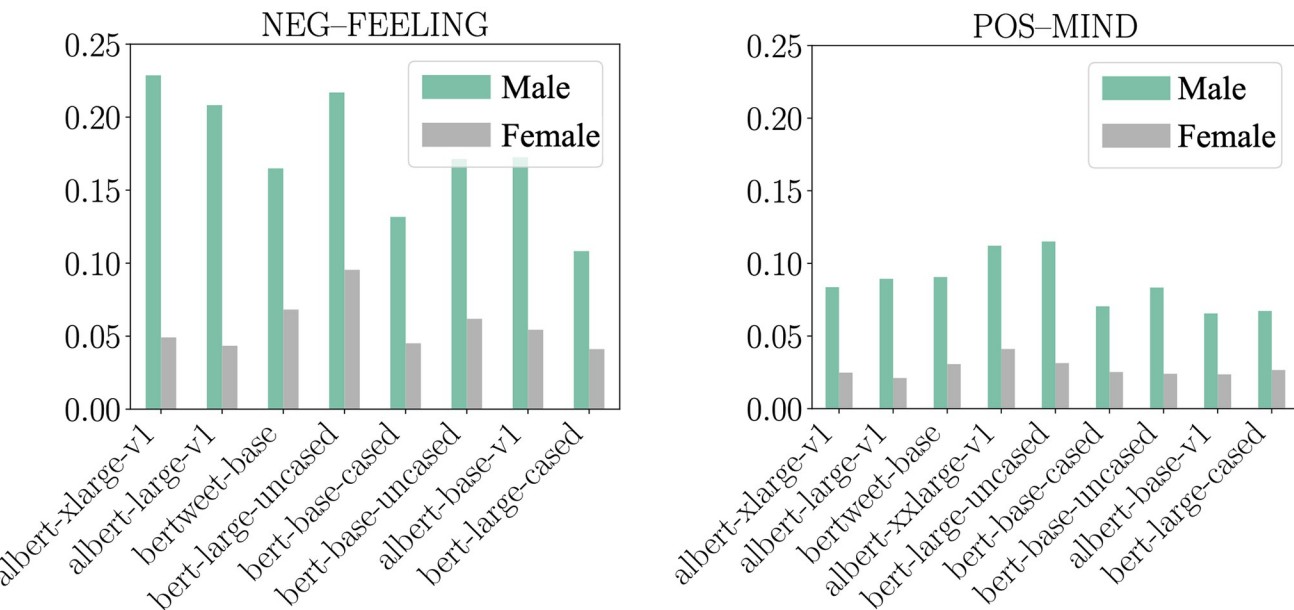

**Fig 3. The frequency with which the 100 largest-deviation adjectives for male and female gender correspond to the supersense "feeling" for the negative sentiment and the supersense "mind" for the positive sentiment.** Results presented for language models with significant differences ($p < 0.004$) between male and female politicians after Bonferroni correction for the number of supersenses (here, 13).

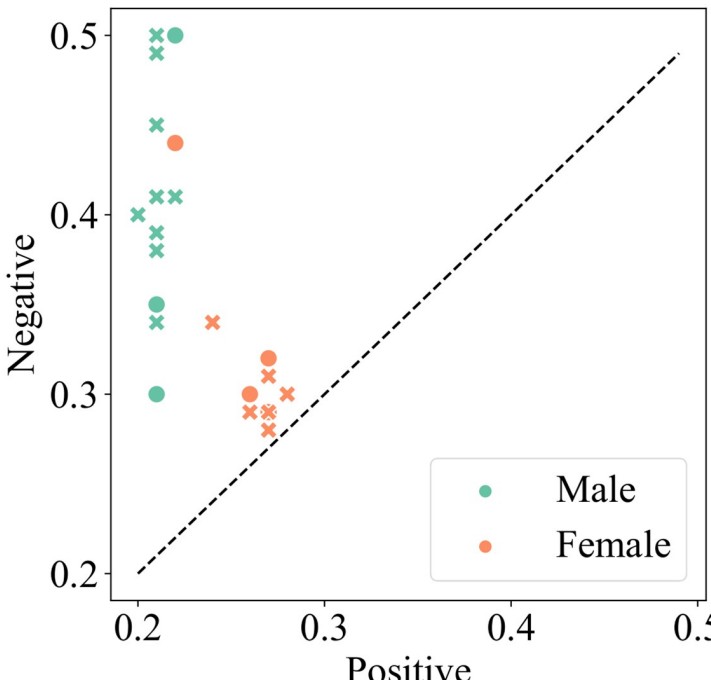

**Fig 4. Mean frequency with which the 100 largest-deviation adjectives for male and female genders correspond to positive or negative sentiment in English.** Each point denotes a language model. Significant differences ($p < 0.05$) are represented with 'x' markers.

sentiment in the descriptions of male politicians generated by models of different architectures. We note that since we are not able to separate the effects of model design and training data, the term architecture encompasses both aspects of pre-trained language models. In particular, XLNet tends to generate more words of negative sentiment compared to other models examined. Surprisingly, larger models tend to exhibit similar gender biases to smaller ones.

**Table 5. ANOVA computed group mean sentiment for male and female genders for adjectives generated by monolingual language models.** Significant differences ($p < 0.05$) are indicated in bold and the dashes denote a baseline group for the analyzed parameter.

| Parameter | Male | | | Female | | |
|---|---|---|---|---|---|---|
| | *neg* | *neu* | *pos* | *neg* | *neu* | *pos* |
| Intercept | **0.390** | **0.401** | **0.209** | **0.284** | **0.435** | **0.281** |
| *Model architecture* | | | | | | |
| ALBERT | — | — | — | — | — | — |
| BERT | -0.016 | 0.014 | 0.001 | -0.006 | 0.016 | **-0.010** |
| BERTweet | **0.057** | **0.005** | 0.004 | 0.008 | 0.002 | -0.010 |
| RoBERTa | **-0.092** | **0.088** | 0.005 | 0.001 | 0.006 | -0.007 |
| XLNet | **0.107** | **-0.114** | 0.007 | **0.086** | **-0.040** | **-0.046** |
| *Model size* | | | | | | |
| base | — | — | — | — | — | — |
| large | 0.000 | -0.002 | -0.002 | **0.035** | **-0.028** | -0.008 |
| xlarge | 0.022 | -0.022 | 0.000 | -0.004 | 0.010 | -0.006 |
| xxlarge | **0.104** | **-0.108** | 0.004 | 0.012 | 0.005 | **-0.017** |
| *p*-value | 0.00 | 0.00 | 0.14 | 0.00 | 0.00 | 0.00 |

## 5.2 Multilingual setup

**5.2.1 PMI and latent-variable model.**   For PMI scores, a pattern similar to the monolingual setup holds. Words associated with female politicians often relate to their appearance and social characteristics such as *beautiful* and *sweet* (prevalent for English, French, and Chinese) or *attentive* (in Russian), whereas male politicians are described as *knowledgeable*, *serious*, or (in Arabic) *prophetic*. Again, we were not able to detect any patterns in words generated around politicians of non-binary gender, where generated words vary from *similar* and *common* (as in French and Russian) or *angry* and *unique* (as in Chinese).

The results of the latent-variable model confirm the previous findings (for an example, see Table 6 for Spanish). Some of the more male-skewed words such as *dead*, and *designated* are still often associated with female politicians given the relatively low deviation scores. Words of positive sentiment used to describe male politicians are often *successful* (Arabic), or *rich* (Arabic, Russian). In a negative context, male politicians are described as *difficult* (Chinese and Russian) or *serious* (prevalent in French and Hindi), and the associated verbs are *sentence* (in Chinese) and *arrest* (in Russian). Notably, words generated in Russian have a strong negative connotation such as *criminal* and *evil*. Positive words associated with female politicians are mostly related to their appearance, while there is no such pattern for words of negative sentiment.

Unlike for English, we did not have access to pre-defined lists of supersenses in the multilingual scenario. We therefore analyzed word representations of the generated words and resorted to cluster analysis to identify semantic groups among the generated adjectives. For each of the generated words, we extracted their word representations using the respective language model. We then performed a cluster analysis for each of the languages and language models analyzed, using the *k*-means clustering algorithm on the extracted word representations. We conducted this analysis separately for each gender to analyze differences in clusters generated for different genders. In each gender–language pair there are clusters with words describing nationalities such as *basque* and *arabe* in French (see Table 7 and S4 Table. Cluster Analysis in the Appendix). Furthermore, regardless of language, there are clusters of words typically associated with the female gender, such as *beautiful*. The distribution of genders for which the words were generated in each cluster is relatively equal across all clusters. Fig 5 shows the distribution of genders for which the words were generated for Arabic with the XLM-base model. These results are valid in all languages and

**Table 6. The top 10 adjectives, for female and male politicians, that have the largest average deviation for each sentiment, extracted from all multilingual models for Spanish.**

| female | | | | | | male | | | | | |
|---|---|---|---|---|---|---|---|---|---|---|---|
| negative | | neuter | | positive | | negative | | neuter | | positive | |
| infantil | 1.9 | embarazado | 1.8 | paciente | 2.2 | destruir | 1.0 | especialista | 0.6 | gratis | 1.2 |
| rival | 1.9 | urbano | 1.8 | activo | 2.1 | cruel | 1.0 | editado | 0.6 | emprendedor | 1.2 |
| chica | 1.9 | único | 1.8 | dulce | 2.0 | peor | 1.0 | izado | 0.6 | extraordinario | 1.2 |
| fundadora | 1.9 | mágico | 1.8 | brillante | 2.0 | imposible | 1.0 | enterrado | 0.6 | defender | 1.2 |
| frío | 1.9 | acusado | 1.8 | amiga | 2.0 | vulgar | 1.0 | cierto | 0.6 | paciente | 1.2 |
| asesino | 1.8 | pintado | 1.8 | óptimo | 1.9 | muerto | 0.9 | incluido | 0.6 | mejor | 1.1 |
| protegida | 1.8 | doméstico | 1.8 | informativo | 1.9 | irregular | 0.9 | denominado | 0.6 | apropiado | 1.1 |
| biológico | 1.8 | crónico | 1.8 | bonito | 1.9 | enfermo | 0.9 | escrito | 0.6 | superior | 1.1 |
| invisible | 1.8 | dominado | 1.8 | mejor | 1.9 | ciego | 0.9 | designado | 0.6 | espectacular | 1.1 |
| magnético | 1.8 | femenino | 1.8 | dicho | 1.9 | enemigo | 0.9 | militar | 0.6 | excelente | 1.1 |

**Table 7. Results of the cluster analysis for French for words generated with m-BERT in association with male politicians.** We list 5 words from every cluster.

| Cluster | Example words |
|---|---|
| 1 | catholique, ancien, petit, bien |
| 2 | premier, international, mondial, directeur |
| 3 | roman, basque, normand, clair, baptiste |
| 4 | franc, arabe, italien, turc, serbe |
| 5 | rouge, blanc, noir, clair, vivant |

language models. However, based on our previous latent-variable model's results, words associated with male politicians are also often used to describe female politicians. The same is not true for female-biased words, which do not appear as often when describing male politicians.

**5.2.2 Sentiment analysis.** We additionally analyzed the overall sentiment of the six cross-lingual language models towards male and female politicians for the selected languages. Our analysis suggests that sentiment towards politicians varies depending on the language used. For English, female politicians tend to be described more positively as opposed to Arabic, French, Hindi, and Spanish. For Russian, words associated with female politicians are more polarized, having both more positive and negative sentiments. No significant patterns for Chinese were detected. See Fig 6 for details.

Finally, analogously to the monolingual setup, we investigated whether there were any significant differences in sentiment dependent on the target language, language model sizes, and architectures; see ANOVA analysis in Table 8. Both XLM and XLM-RoBERTa generated fewer negative and more positive words than BERT multilingual, *e.g.*, the mean frequency with which the 100 largest deviation adjectives for the male gender correspond to negative sentiment is lower by 2.00% and 5.73% for XLM and XLM-R, respectively. Indeed, as suggested above, we found that language was a highly significant factor for bias in cross-lingual language models, along with model architecture. For English and French, *e.g.*, generated words were

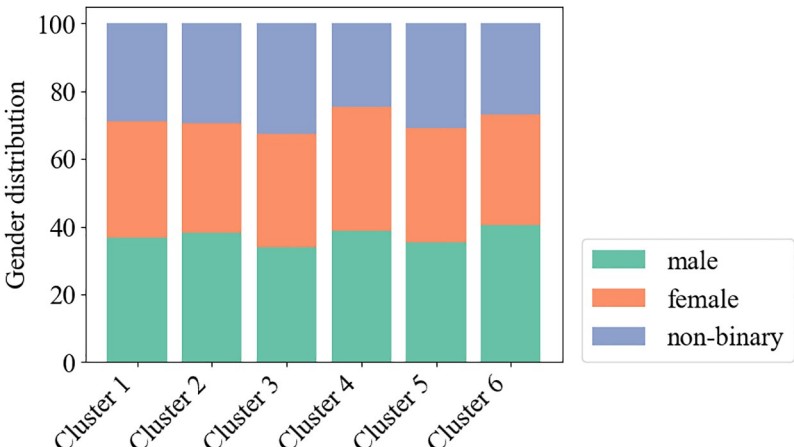

**Fig 5. Distribution of genders in each cluster identified within word representations generated for Arabic the XLM-base language model.**

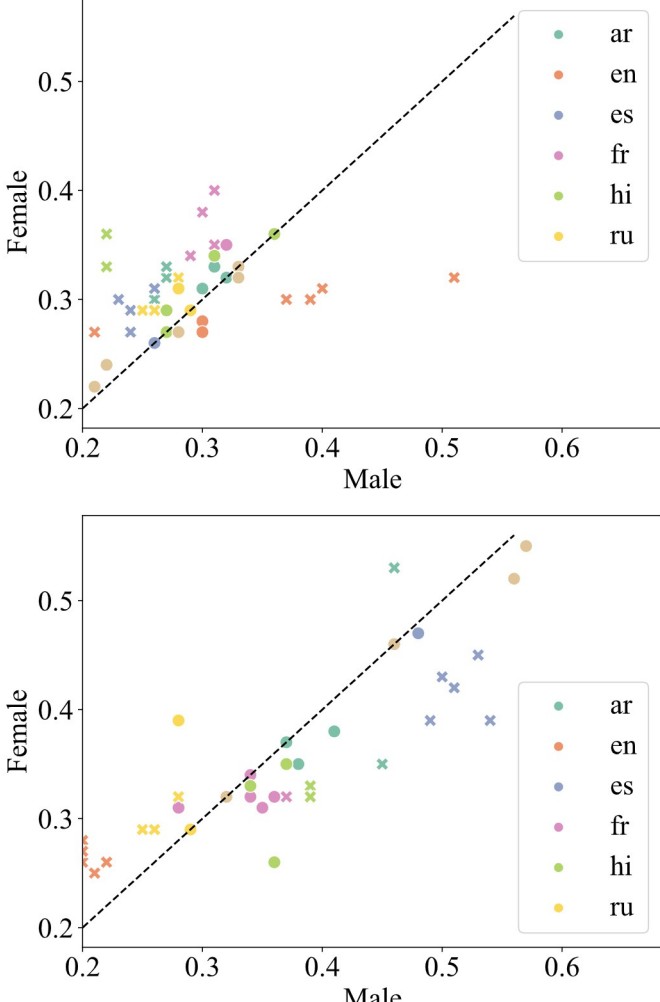

**Fig 6. Mean frequency with which the top 100 adjectives–the most strongly associated with either male or female gender–correspond to negative (top) and positive (bottom) sentiment.** Significant differences ($p < 0.05$) are represented with 'x' markers.

often more negative when used to describe male politicians. Surprisingly, we did not observe a significant influence of model size on the encoded bias.

## 6 Limitations

### 6.1 Potential harms in using gender-biased language models

Prior research has unveiled the prevalence of gender bias in political discourse, which can be picked up by NLP systems if trained on such texts. Gender bias encoded in large language models is particularly problematic, as they are used as the building blocks of most modern NLP models. Biases in such language models can lead to gender-biased predictions, and thus reinforce harmful stereotypes extant in natural language when these models are deployed. However, it is important to clarify that by our definition, while bias does not have to be harmful (*e.g.*, *female* and *pregnant* will naturally have a high PMI score) [73], it might be in several instances (*e.g.*, a positive PMI between *female* and *fragile*).

**Table 8. ANOVA computed group mean sentiment values for male and female genders for adjectives generated by cross-lingual language models.** Significant differences ($p < 0.05$) are in bold and the dashes denote a baseline group for the analyzed parameter.

| Parameter | Male | | | Female | | |
|---|---|---|---|---|---|---|
| | *neg* | *neu* | *pos* | *neg* | *neu* | *pos* |
| Intercept | **0.310** | **0.289** | **0.401** | **0.327** | **0.326** | **0.347** |
| *Model architecture* | | | | | | |
| m-BERT | — | — | — | — | — | — |
| XLM | **-0.020** | **-0.019** | **0.039** | **-0.013** | 0.0006 | **0.025** |
| XLM-RoBERTa | **-0.057** | 0.008 | **0.050** | **-0.014** | -0.005 | **0.029** |
| *Model size* | | | | | | |
| base | — | — | — | — | — | — |
| large | 0.007 | 0.006 | **-0.013** | 0.002 | -0.003 | -0.006 |
| *Language* | | | | | | |
| Arabic | — | — | — | — | — | — |
| Chinese | **-0.031** | **-0.032** | **0.063** | **-0.056** | **-0.049** | **0.106** |
| English | **0.091** | **0.126** | **-0.216** | **-0.025** | **0.125** | **-0.100** |
| French | **0.024** | **0.052** | **-0.077** | **0.062** | **-0.026** | **-0.350** |
| Hindi | -0.011 | **0.091** | **-0.080** | **0.019** | **0.026** | **-0.044** |
| Russian | -0.016 | **0.173** | **-0.156** | **-0.023** | **0.093** | **-0.069** |
| Spanish | **-0.031** | **-0.041** | **0.081** | **-0.033** | **-0.033** | **0.066** |
| *p*-value | 0.00 | 0.00 | 0.00 | 0.00 | 0.00 | 0.00 |

## 6.2 Quality of collaborative knowledge bases

For the purpose of this research, it is imperative to acknowledge the presence of gender bias in Wikipedia, which is characterized by a clear disparity in the number of female editors [74], a smaller percentage of notable women having their own Wikipedia page, and these pages being less extensive [75]. Indeed, we observe this disparity in the gender distribution in Table 1. We gathered information on politicians from the open-knowledge base Wikidata, which claims to do gender modeling at scale, globally, for every language and culture with more data and coverage than any other resource [76]. It is a collaboratively edited data source, and so, in theory, everyone could make changes to an entry (including the person the entry is about), which poses a potential source of bias. Since we are only interested in overall gender bias trends as opposed to results for individual entities, we can tolerate a small amount of noise.

## 6.3 Gender selection

In our analysis, we aimed to incorporate genders beyond male and female while maintaining statistical significance. However, politicians of non-binary gender cover only 0.025% of collected entities. Further, politicians with no explicit gender annotation were not considered in our analysis. Furthermore, it is plausible that this set could be biased towards non-binary-gendered politicians. This restricts possible analyses for politicians of non-binary gender and risks drawing wrong conclusions. Although our method can be applied to any named entities of non-binary gender to analyze the stance towards them, we hope future work will obtain more data on politicians of non-binary gender to avoid this limitation and to enable a fine-grained study of gender bias towards diverse gender identities departing from the categorical view on gender.

## 6.4 Beyond English

We explored gender bias encoded in cross-lingual language models in seven typologically distinct languages. We acknowledge that the selection of these languages may introduce

additional biases to our study. Further, the words generated by a language model can also simply reflect how particular politicians are perceived in these languages, and how much they are discussed in general, rather than a more pervasive gender bias against them. However, considering our results in aggregate, it is likely that the findings capture general trends of gender bias. Finally, a potential bias in our study may be associated with racial biases that are reflected by a language model, as names often carry information about a politician's country of origin and ethnic background.

## 7 Conclusions

In this paper, we have presented the largest study of quantifying gender bias towards politicians in language models to date, considering a total number of 250k politicians. We established a novel method to generate a multilingual dataset to measure gender bias towards entities. We studied the qualitative differences in language models' word choices and analyzed sentiments of generated words in conjunction with gender using a latent-variable model. Our results demonstrate that the stance towards politicians in pre-trained models is highly dependent on the language used. Finally, contrary to previous findings [14], our study suggests that larger language models do not tend to be significantly more gender-biased than smaller ones.

While we restricted our analysis to seven typologically diverse languages, as well as to politicians, our method can be employed to analyze gender bias towards any NEs and in any language, provided that gender information for those entities is available. Future work will focus on extending this analysis to investigate gender bias in a wider number of languages and will study this bias' societal implications from the perspective of political science.

## Supporting information

**S1 Text. Politician gender.**
(PDF)

**S2 Text. Supersenses.**
(PDF)

**S3 Text. Additional experiments.**
(PDF)

**S1 Table. Word orderings.**
(PDF)

**S2 Table. Sentiment analysis evaluation.**
(PDF)

**S3 Table. Generated words.**
(PDF)

**S4 Table. Cluster analysis.**
(PDF)

## Acknowledgments

The authors would like to thank Eleanor Chodroff, Clara Meister, and Zeerak Talat for their feedback on the manuscript.

## Author Contributions

**Conceptualization:** Karolina Stańczak, Sagnik Ray Choudhury, Tiago Pimentel, Ryan Cotterell, Isabelle Augenstein.

**Data curation:** Karolina Stańczak, Sagnik Ray Choudhury, Tiago Pimentel.

**Formal analysis:** Karolina Stańczak, Sagnik Ray Choudhury, Tiago Pimentel.

**Funding acquisition:** Ryan Cotterell, Isabelle Augenstein.

**Investigation:** Karolina Stańczak, Sagnik Ray Choudhury, Tiago Pimentel.

**Methodology:** Karolina Stańczak, Sagnik Ray Choudhury, Tiago Pimentel.

**Resources:** Karolina Stańczak, Sagnik Ray Choudhury, Tiago Pimentel.

**Software:** Karolina Stańczak, Sagnik Ray Choudhury, Tiago Pimentel.

**Supervision:** Ryan Cotterell, Isabelle Augenstein.

**Validation:** Karolina Stańczak, Sagnik Ray Choudhury, Tiago Pimentel.

**Visualization:** Karolina Stańczak, Sagnik Ray Choudhury, Tiago Pimentel.

**Writing – original draft:** Karolina Stańczak, Sagnik Ray Choudhury, Tiago Pimentel.

**Writing – review & editing:** Karolina Stańczak, Sagnik Ray Choudhury, Tiago Pimentel.

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
