## [Decision Letter · Decision Letter 0]

21 Apr 2022

PONE-D-22-03986Quantifying Gender Bias Towards Politicians in Cross-Lingual Language ModelsPLOS ONE

Dear Dr. Stanczak,

Thank you for submitting your manuscript to PLOS ONE. After careful consideration, we feel that it has merit but does not fully meet PLOS ONE’s publication criteria as it currently stands. Therefore, we invite you to submit a revised version of the manuscript that addresses the points raised during the review process.

We look forward to receiving your revised manuscript.

Kind regards,

Yang Xu, PhD

Academic Editor

PLOS ONE

Journal Requirements:

“This work is partly funded by Independent Research Fund Denmark under grant

agreement number 9130-00092B.”

“This work is mostly funded by Independent Research Fund Denmark under grant agreement number 9130-00092B.”

“This work is partly funded by Independent Research Fund Denmark under grant

agreement number 9130-00092B.”

Additional Editor Comments:

This paper provides a computational analysis of gender bias toward politicians in language models. Three expert reviewers have offered detailed comments on the manuscript. Although overall the reviewers agreed that the authors have made an attempt at a valuable problem, they pointed out many issues in the current version (most prominently, the lack of thoroughness in the exposition, and the extension and deepening of analyses) and provided extensive feedback on how to improve the manuscript in a major revision. The manuscript as it stands is not acceptable for publication at PLOS, and I encourage the authors to address the reviewer comments carefully in the revision.

R1 raises an issue about the input sources of the language models and controlling for the input. R1 also suggests a controlled analysis based on temporal considerations and raises a question about the discrepancy between English and other languages in the top adjectives generated.

R2 raises a concern about the choice and relevance of politician names and suggests separate analyses to be performed on subsets of the names, tailored for individual languages. R2 and R1 indicate on the reviewer feedback form that the data underlying the findings have not been made fully available, which is contradictory to the authors’ indication on the submission form — if this is a misunderstanding, please indicate clearly where the data can be accessed in the manuscript (and in the forthcoming response letter). It is also advised that the authors provide the link to the code repository that replicates all the analyses reported in the manuscript to be revised.

R3 suggests that the writing needs be substantially improved in terms of its structure, justification for the methodological choices (see R2’s concurrent comment point #3., and R1’s final comment), and correction for an overstatement about novel methodologies. R3 also points out correctly that the results presented seem surface explorations, and more in-depth analyses should be considered to address this issue in the cross-linguistic setting which include cross-language variability, reliability of methodology across languages and politicians. R2 concurs with this comment suggesting that the paper presented similar methodologies to an existing publication on gender word associates, but more analyses should be developed across languages, as well as those that provide better interpretation of the results and some correlational analyses on term frequencies.

Reviewers' comments:

Reviewer's Responses to Questions

**Comments to the Author**

1. Is the manuscript technically sound, and do the data support the conclusions?

Reviewer #1: Yes

Reviewer #2: Yes

Reviewer #3: Partly

2. Has the statistical analysis been performed appropriately and rigorously? 

Reviewer #1: Yes

Reviewer #2: Yes

Reviewer #3: Yes

3. Have the authors made all data underlying the findings in their manuscript fully available?

Reviewer #1: No

Reviewer #2: No

Reviewer #3: Yes

4. Is the manuscript presented in an intelligible fashion and written in standard English?

Reviewer #1: Yes

Reviewer #2: Yes

Reviewer #3: No

5. Review Comments to the Author

Reviewer #1: The submitted work investigates the inherent gender biases learned by large contextualized neural language models (LMs) by probing sentiment biases of words that LMs associate with both male and female politicians. Distinct from previous work that attempts to quantify gender biases in language models, the authors consider a multi-lingual setting where both monolingual LMs and cross-lingual LMs are analyzed. Combined with previous methods in quantifying gender bias (PMI and latent variable model from Hoyle et al. (2019)), the authors propose a novel experimental setup that considers lexical sentiment scores of words that are predicted by the LMs to directly supersede or following the named entity of interest. Under the proposed experimental setup, gender biases in LMs are quantified by categorizing the list of words with the largest deviation w.r.t. gender, as well as statistical tests w.r.t. hyperparameters such as model size, model architecture, and language. The proposed methodology and experiment setup has the potential to be extended to analyze gender biases toward other types of named entities.

The authors promise to make the data fully available but I was not able to find the data in the materials submitted.

Detailed questions and comments regarding the submission:

- What datasets are the language models pre-trained on? It is conceivable that the data sources themselves could be inherently biased in different ways (e.g. different news agencies having different biases). When evaluating the effect of specific model architecture on biases captured, it may be worthwhile to consider a more controlled setting where all models are trained on the same data source.

- Have you controlled for the time period in which the politicians are active? If so, which time periods do you consider? Not only that gender biases change over time, but also political sentiment in general. These changes can confound your result where some sentiment differences are not necessarily caused by gender differences.

- The top 100 generated adjectives are considered for all languages but only the top 20 adjectives are considered for English. Is there a reason for this discrepancy?

- In Table 4, what are the number of parameters associated with the model sizes (i.e large, xlarge, xxlarge)? Are these sizes consistent across the different architectures in terms of parameters counts?

Reviewer #2: The paper describes the problem of gender bias in language models and conducts a cross-lingual study to identify it. The gender-bias in language models is proxied through politicians' names and sentiment lexicon. The main claim is straightforward. The degree of gender bias is different in each language, and larger language models are not significantly more biased than smaller ones.

The problem studied in this paper has fundamental social implications and the paper has an interesting point of view. However, I have some concerns that require revision.

1) My main concern is about the choice of politicians' names. According to the paper, 250k names are retrieved from Wikidata. It is assumed that because these 250k names are for politicians and the fact that politicians have higher toxicity associations and bias in NLP models (Prabhakaran et al.), then the chosen names should represent the embedded gender bias in language models. However, I believe that not all of the names are relevant in all the 7 languages studied in the paper.

Although the BERT-based models are trained on Wikipedia, I doubt that a translated article about a politician on Wikipedia is enough to impose gender bias on the model. I strongly suggest that the experiments be run separately on a smaller set of names specific to each language to reduce the noise (e.g., for the Chinese language, only experiment on the Chinese politicians + a few internationally well-known politicians).

2) Most of the experiments in the paper are done based on the frequency with which the top K words associated with a gender belong to a category (e.g., Negative sentiments, Positive sentiments, or a supersense). Although the results are interesting and should be reported, they are very similar to the ones in the previous work of Hoyle et al., 2019, except that now they are done cross-lingually and based on the politicians' data. I believe more experiments and tests should be done to assess gender bias and its differences across languages and language models. Exploring questions like which supersenses are more/less responsible for gender bias in different languages/LMs, and how systematically similar or different are the top gender-associated adjectives across languages can further strengthen the findings.

3) I faced some problems understanding section 5.1.2 and more specifically figure 3. How are you deriving the 100 largest-deviation adjectives for male and female genders? Isn't \\eta(w, s) used for the deviation metric? If so how are you marginalizing the sentiment in this function? I think adding some detail about deriving the frequency of largest-deviation adjectives helps clarify this section.

4) The paper proposes a critical point about the gendered adjectives and neutral words, and how they might mitigate the bias in language models trained on gendered languages (e.g., in Spanish). I am curious if there are ways to examine this proposal against the biases detected in the LMs corresponding to these languages. Some correlation analyses between the frequency of feminine/masculine inanimate words and the frequency of negative/positive adjectives associated with each gender might be one simple approach to this.

My general argument is that the methodology used in the paper is statistically sound and evaluated in previous work, and the findings support the claim (although the claims themselves are not impressive), however, the experiments should be extended to deliver a more extensive set of findings.

Reviewer #3: This paper explores gender bias toward politicians in transformer language models in a cross-lingual scenario. In particular, it investigates the adjectives/verbs associated with a large number of politicians in 7 languages, and measures deviations in sentiment toward male/female/'other' politicians in the respective language groups.

While the motivation and contribution of data set is laudable, as is the treatment of gender as non-binary. However, the paper (1) is not well-structured, at times repetitive and difficult to follow (esp for non-domain expert readers as attracted in PLOSONE); and (2) lacks motivation for several methodological decisions. Finally, the paper abstract/intro suggests a methodological contribution, while in section 4.2 it becomes clear that it simply re-uses a previously proposed method; and this method is not even described in detail.

In sum, I think this paper can make a worthwhile contribution by studying an important phenomenon at a broad scale, however, it needs careful rewriting and very careful motivation and validation of all steps of the approach. Please find detailed comments/suggestions below.

Comments on paper quality / structure

- Section 3 dives right into templated probing methodology without motivating this approach. This compromises understandability of the paper, especially for an audience not focused on NLP (like PLOSONE). I think the paper would be more readable if Related Work was presented earlier on.

- Similarly, for section 3.3. I think an illustration early on in the paper, of the overall approach (and goal) would help.

- Information is often presented in confusing order, e.g., a long list of models used (lines 133ff.) interrupting the explanation of generation methodology/POS filtering.

- The experiment setup is incomprehensible. A paper should be self-contained and "we conduct the same analysis as presented in [12]" (l. 244) is not sufficient.

Comments on methodology

- A discussion of the type of bias targeted would help, in particular (as far as I understand) it addresses bias based on properties of the *speaker* (i.e., language user) not bias toward properties of the entity being *spoken about* (i.e,. the politician). These dimensions have been disentangled and addressed separately (and jointly) in prior work (e.g., [1]).

- The exact method of retrieving politicians names remains unclear. Was the process quality-checked? How well did it work for each language? How do you align politicians' names across languages?

- The generation method still relies on syntactic cues, and will work better for languages in which the required bigrams are more plausible. E.g., for EN presumably all adjectives and most verbs will have very low probability. To what extent is the ranking meaningful? Was there any quality analysis across language?

- Please motivate all parameters, including choice of languages, cut-off parameters (show the top20 / top100 words, rather than counts only)

- An ablation study on comparable names (e.g., same number of subword tokens, comparable popularity) would have been interesting to disentangle LM perplexity from actual bias effects.

- Similarly, the approach for creating sentiment lexica seems highly complicated. How did you choose the base models? Is there any evidence for the hypothesis that "combining a larger number of individual lexica with SentiVAE leads to more reliable results" (l. 173), i.e., how well did the most competitive base model perform in the tasks in Table 10?

- Table 10 does suggest significant drops for Chinese, Hindi and French; as well as a very low overall performance for Spanish. The conclusion that "the

performance of over 72% is satisfactory and validates the lexicon approach" seems premature given that the bias detection method relies on quality sentiment classification, and the unconvincing results in Table 10.

- Line 219 treats gender as binary, while the data set seems to be more nuanced (m, f, other). Please explain / unify.

Comments on results / analyses

- Given that the contribution is a *cross-lingual* analysis, the corresponding results section is very shallow, the main take away being relative bias of small vs larger LMs.

- There's a missed opportunity of exploring in-depth research questions/analyses such as: are there systematic differences in bias across languages? How reliable is the method across languages (incl name selection, data generation, bias measurement)? How reliable is the method for high/low coverage politicians (potentially including politicians of the same vs. different native language).

[1] https://aclanthology.org/2020.emnlp-main.23.pdf

6. PLOS authors have the option to publish the peer review history of their article (what does this mean?). If published, this will include your full peer review and any attached files.

Reviewer #1: No

Reviewer #2: No

Reviewer #3: No

---

## [Author Response · Author response to Decision Letter 0]

22 Jun 2022

Dear PLOS ONE Academic Editor,

We would like to start by thanking the reviewers for their time and consideration in evaluating our work, and we are writing to ask you to examine a revised version of our manuscript. As a summary, our paper explores a simple method to probe pre-trained language models for gender bias, which we use to effect a multi-lingual study of gender bias towards politicians. We construct a dataset of 250k politicians from most countries in the world and quantify adjective and verb usage around those politicians’ names as a function of their gender. We conduct our study in 7 languages across 6 different language modeling architectures.

In the revised version of the paper, we incorporate the reviewers’ feedback and add additional experiments, where we conduct a more in-depth analysis of gender bias towards politicians in multilingual pre-trained language models. We extend our experiments with three additional experimental setups: 1) We perform an analysis on a subset of the most popular politicians for each of the analyzed languages; 2) We conduct an analysis on a subset of names where for each language we examine words generated for politicians whose country of origin (i.e., their nationality) uses the respective language as an official language; 3) We split our dataset and analyze words generated for politicians born before and after Baby Boom (1946). We believe that these experiments enable analysis in a more controllable setup. However, we did not find any significant patterns. We include these results in the Appendix, while being open to moving them upon acceptance to the main body of the paper if the reviewers consider it beneficial for the paper.

Following feedback from reviewers, we have restructured the main body of the paper to increase its readability. We have expanded upon our definition of gender bias and type of bias analyzed. We have clarified our data generation process and have included an overview graph. Further, we have extended justification of our methodological choices in the experimental setup. We believe that these changes have increased the clarity of the paper. 

Additionally, we have now made our dataset and code available in order to increase transparency and reproducibility of the paper and the conducted analysis. Since our approach can be extended to other types of entities, biases, and other languages, we hope that the presented methodology will initiate further studies of societal biases in large pre-trained language models. We include the list of politician names, generated words, and our code as an attachment to our submission. 

Finally, we clarified in the attached cover letter that the funders had no role in study design, data collection and analysis, decision to publish, or preparation of the manuscript.

We thank the reviewers for the time and energy put into our submission. We respond to the specific questions the reviews have raised in the full response letter.

---

## [Decision Letter · Decision Letter 1]

8 Aug 2022

PONE-D-22-03986R1Quantifying Gender Bias Towards Politicians in Cross-Lingual Language ModelsPLOS ONE

Dear Dr. Stanczak,

Thank you for submitting your manuscript to PLOS ONE. After careful consideration, we feel that it has merit but does not fully meet PLOS ONE’s publication criteria as it currently stands. Therefore, we invite you to submit a revised version of the manuscript that addresses the points raised during the review process.

We look forward to receiving your revised manuscript.

Kind regards,

Yang Xu, PhD

Academic Editor

PLOS ONE

Additional Editor Comments:

Two reviewers provided comments on the revised manuscript. Although R1 found the paper acceptable, R2 raised a series of issues that should be addressed and discussed. Specifically, R2 suggested an in-depth analysis on the semantic representations of language models (that could have given rise to the results), a clearer explanation of the methodology in Sec 5.2 (with the possible addition of an illustrative figure) and the working of SentiVAE in Sec 4.3, a more rigorous evaluation in Sec 4.3 e.g., using potentially human annotations and ablation, and a more detailed description of the supersets in Sec 6.1 and 6.1.2. These are reasonable comments, and I look forward to reading  the next revision.

Reviewers' comments:

Reviewer's Responses to Questions

**Comments to the Author**

1. If the authors have adequately addressed your comments raised in a previous round of review and you feel that this manuscript is now acceptable for publication, you may indicate that here to bypass the “Comments to the Author” section, enter your conflict of interest statement in the “Confidential to Editor” section, and submit your "Accept" recommendation.

Reviewer #1: All comments have been addressed

Reviewer #2: (No Response)

2. Is the manuscript technically sound, and do the data support the conclusions?

Reviewer #1: Yes

Reviewer #2: Yes

3. Has the statistical analysis been performed appropriately and rigorously? 

Reviewer #1: Yes

Reviewer #2: Yes

4. Have the authors made all data underlying the findings in their manuscript fully available?

Reviewer #1: Yes

Reviewer #2: Yes

5. Is the manuscript presented in an intelligible fashion and written in standard English?

Reviewer #1: Yes

Reviewer #2: No

6. Review Comments to the Author

Reviewer #1: The authors have adequately addressed comments raised in my previous review and have specifically improved clarity on their experimental setup as well as motivations behind various experimental decisions. An additional experiment has been performed to investigate the temporal effect of political sentiment.

To further improve the paper’s clarity, it may be worthwhile to quickly elaborate on the decision to only include 20 adjectives in the main paper (e.g. selecting top 20 adjectives in English already offers a larger pool of adjectives) similar to what has been provided in the response letter.

A sample of the dataset used has now been provided and can be accessed. The authors promise the public release of their data upon publication.

Reviewer #2: The paper describes the problem of gender bias in language models and conducts a cross-lingual study to identify it. The gender bias in language models is proxied through politicians' names and sentiment lexicon. The main claim is straightforward. The degree of gender bias is different in each language, and larger language models are not significantly more biased than smaller ones.

Regarding my previous comments, the authors included new experiments for different sets of politician names. The results were like that of the initial experiment. However, I still believe that the paper lacks a more in-depth analysis. The current experiments show that language modes suffer from gender bias toward politicians, but this bias does not depend on the size of the model but on the language itself. Investigating how the representation of these languages varies in the semantic space generated by the language model could explain the current results.

The methodology is almost completely borrowed from Hoyle et al., 2013.

The paper, however, does not clearly explain how this method works in section 5.2. The current description is rather incomplete and difficult to follow without reading Hoyle et al., especially for those unfamiliar with NLP methodologies. I recommend rewriting this section to increase its readability. Adding an illustrative figure to summarize the approach would be helpful as well.

The same thing applies to section 4.3. How does SentiVAE work? What are the differences in the chosen sentiment lexica? The generation of the multilingual sentiment lexica is a valid and important contribution of the paper and should be emphasized, but only a few paragraphs are dedicated to explain the approach and the results. Also, a more rigorous evaluation is needed for this section, especially for Hindi and Spanish, as the F1 scores are lower than the rest of the languages. Currently, the authors evaluate the generated sentiment lexicon based on the average sentiment of words in a sentence. Having a token-level approach toward sentiment analysis would indeed generate some noise. Using human annotations (even for a small section of the words) or conducting correlation analysis with the initial sentiment Lexia might be more accurate.

As an ablation study, the authors could repeat the experiments in section 6 with one of the monolingual sentiment Lexia. This experiment clarifies whether using SentiVAE helps to identify gender bias in the language models or not.

I still recommend the authors to provide a more in-depth analysis that would go beyond the experiments and the results of the previous studies. The paper studies an important question and the authors have collected valuable data to investigate the question, but the experiments and the results provide only a surface-level analysis of this problem.

The readability of some parts, especially the methodology and the experiments, should be improved as well.

Writing comments:

In section 6.1., the authors could provide a description of the supersets used and the set of words each contains.

I find it difficult to understand the first paragraph of section 6.1.2. Could you provide some examples for each step mentioned in this section?

7. PLOS authors have the option to publish the peer review history of their article (what does this mean?). If published, this will include your full peer review and any attached files.

Reviewer #1: No

Reviewer #2: No

---

## [Author Response · Author response to Decision Letter 1]

22 Sep 2022

Dear PLOS ONE Academic Editor,

We would like to start by thanking the reviewers for their time and consideration in evaluating our work, and we are writing to ask you to examine a revised version of our manuscript. 

As a summary, our paper explores a simple method to probe pre-trained language models for gender bias, which we use to effect a multi-lingual study of gender bias towards politicians. We construct a dataset of 250k politicians from most countries in the world and quantify adjective and verb usage around those politicians’ names as a function of their gender. We conduct our study in 7 languages across 6 different language modeling architectures.

We have considered reviewer 2’s feedback and have extended the experimental section of the paper. We have conducted a more in-depth analysis of the words generated for the analyzed languages and language models in order to investigate semantic spaces in which words generated by language models are represented. In particular, we analyze word embeddings of the generated words for each of the three distinguished genders. To this end, we have employed PCA and a cluster analysis in order to identify semantic groups of generated words. We find that often the same words are being generated in association with different genders. However, based on previous results, words typically associated with the female gender are indeed generated more often to describe female politicians. 

Further, we have expanded on Sections 4.3, 5.2, and 6.1-6.2 so that the paper is easier to follow for a broader type of audience than the NLP community. We have added more examples to clarify described steps of the analysis. We believe that these changes have clarified the methodology we have used.

We agree that the reliability of our experiments strongly relies on the quality of the sentiment lexica employed. Therefore, as mentioned above, we have extended Section 4.3 to include a more detailed explanation of the algorithm and our reasoning for using it in our analysis. The selected set of lexica that we used are all publicly available and have previously been verified on a sentiment classification task. Ultimately, we want to maximize the coverage of the words while retaining high quality of the lexicon used. However, due to disparities in their scales, we resort to the SentiVAE method to combine them. In the initial experiments, we have tested different combinations of lexica, evaluating them on the listed sentiment analysis datasets. Based on the results of this analysis, we exclude multilingual lexica for Hindi (described in the manuscript), as mentioned in the paper. The lexicon evaluation results for Spanish are indeed better than the results of a state-of-the-art supervised algorithm. Hence, although the results might seem relatively low, we can assume that they deliver results of as high a quality as it is currently possible. Further, the previously curated lexica were already created using human annotations and therefore, we do not see the added value in using more human annotations of the data. SentiVAE is solely a transformation of the lexica from their initial scale to be represented as a vector with values for the three sentiments, positive, negative and neutral. 

In the manuscript, the sections marked in blue indicate revisions we have made for this submission, while red indicates the changes made for the previous one. Below, we respond to the specific questions the reviews have raised inline and we mark our comments with a distinguished font and a different color -- blue. 

PONE-D-22-03986R1

Quantifying Gender Bias Towards Politicians in Cross-Lingual Language Models

PLOS ONE

Dear Dr. Stanczak,

Thank you for submitting your manuscript to PLOS ONE. After careful consideration, we feel that it has merit but does not fully meet PLOS ONE’s publication criteria as it currently stands. Therefore, we invite you to submit a revised version of the manuscript that addresses the points raised during the review process.

We look forward to receiving your revised manuscript.

Kind regards,

Yang Xu, PhD

Academic Editor

PLOS ONE

Additional Editor Comments:

Two reviewers provided comments on the revised manuscript. Although R1 found the paper acceptable, R2 raised a series of issues that should be addressed and discussed. Specifically, R2 suggested an in-depth analysis on the semantic representations of language models (that could have given rise to the results), a clearer explanation of the methodology in Sec 5.2 (with the possible addition of an illustrative figure) and the working of SentiVAE in Sec 4.3, a more rigorous evaluation in Sec 4.3 e.g., using potentially human annotations and ablation, and a more detailed description of the supersets in Sec 6.1 and 6.1.2. These are reasonable comments, and I look forward to reading the next revision.

We thank the reviewers and the editor for their comments and valuable feedback throughout the reviewing process. We thank Reviewer 1 for the positive evaluation. 

We have incorporated Reviewer 2’s feedback by extending the experimental section of the paper. We have also conducted an analysis of the word embeddings of the generated words in order to identify semantic groups across languages and genders.

Further, we have expanded on Sections 4.3, 5.2, and 6.1-6.2, and believe that the resulting paper is clearer for a broader audience beyond NLP. We believe that these changes have clarified the methodology we used.

Since the selected lexica are already based on human annotations, and SentiVAE is solely a data transformation to bring different lexica to the same scale, we did not add human annotations to evaluate the sentiment lexica. We hope that, in light of our explanation, the reviewer would agree that this is not necessary. 

We believe that the changes we have made to the manuscript have increased the readability of the paper and the extended experimental section has added merit to the analysis. 

Reviewers' comments:

Reviewer's Responses to Questions

Comments to the Author

1. If the authors have adequately addressed your comments raised in a previous round of review and you feel that this manuscript is now acceptable for publication, you may indicate that here to bypass the “Comments to the Author” section, enter your conflict of interest statement in the “Confidential to Editor” section, and submit your "Accept" recommendation.

Reviewer #1: All comments have been addressed

Reviewer #2: (No Response)

2. Is the manuscript technically sound, and do the data support the conclusions?

Reviewer #1: Yes

Reviewer #2: Yes

3. Has the statistical analysis been performed appropriately and rigorously?

Reviewer #1: Yes

Reviewer #2: Yes

4. Have the authors made all data underlying the findings in their manuscript fully available?

Reviewer #1: Yes

Reviewer #2: Yes

5. Is the manuscript presented in an intelligible fashion and written in standard English?

Reviewer #1: Yes

Reviewer #2: No

6. Review Comments to the Author

Reviewer #1: The authors have adequately addressed comments raised in my previous review and have specifically improved clarity on their experimental setup as well as motivations behind various experimental decisions. An additional experiment has been performed to investigate the temporal effect of political sentiment.

To further improve the paper’s clarity, it may be worthwhile to quickly elaborate on the decision to only include 20 adjectives in the main paper (e.g. selecting top 20 adjectives in English already offers a larger pool of adjectives) similar to what has been provided in the response letter.

We have now added an explanation to the paper of why we have only chosen 20 adjectives in English as opposed to 100 in other languages. 

A sample of the dataset used has now been provided and can be accessed. The authors promise the public release of their data upon publication.

We thank Reviewer 1 for the helpful comments throughout the process of reviewing our paper and a positive evaluation of the paper in its current form. 

Reviewer #2: The paper describes the problem of gender bias in language models and conducts a cross-lingual study to identify it. The gender bias in language models is proxied through politicians' names and sentiment lexicon. The main claim is straightforward. The degree of gender bias is different in each language, and larger language models are not significantly more biased than smaller ones.

Regarding my previous comments, the authors included new experiments for different sets of politician names. The results were like that of the initial experiment. However, I still believe that the paper lacks a more in-depth analysis. The current experiments show that language modes suffer from gender bias toward politicians, but this bias does not depend on the size of the model but on the language itself. Investigating how the representation of these languages varies in the semantic space generated by the language model could explain the current results.

Following reviewer 2’s feedback, we have added an additional experiment in which we conduct a cluster analysis of words generated for different gender categories in the analyzed languages and language models. These experiments enabled an in-depth analysis of semantic spaces generated by language models, as suggested by reviewer 2. We believe that this analysis has increased the value of the paper. 

The methodology is almost completely borrowed from Hoyle et al., 2013. The paper, however, does not clearly explain how this method works in section 5.2. The current description is rather incomplete and difficult to follow without reading Hoyle et al., especially for those unfamiliar with NLP methodologies. I recommend rewriting this section to increase its readability. Adding an illustrative figure to summarize the approach would be helpful as well.

We have rewritten the methodology section and have elaborated on the model development steps. We believe that these additions clarify the method and make it readable independently of the Hoyle et al. 2019 paper. 

The same thing applies to section 4.3. How does SentiVAE work? What are the differences in the chosen sentiment lexica? The generation of the multilingual sentiment lexica is a valid and important contribution of the paper and should be emphasized, but only a few paragraphs are dedicated to explain the approach and the results. Also, a more rigorous evaluation is needed for this section, especially for Hindi and Spanish, as the F1 scores are lower than the rest of the languages. Currently, the authors evaluate the generated sentiment lexicon based on the average sentiment of words in a sentence. Having a token-level approach toward sentiment analysis would indeed generate some noise. Using human annotations (even for a small section of the words) or conducting correlation analysis with the initial sentiment Lexia might be more accurate.

We thank the reviewer for recognizing the sentiment creation as one of the contributions of our work. We have elaborated now on the SentiVAE method and our choice of lexica in the main body of the paper to highlight the validity of our method. 

As an ablation study, the authors could repeat the experiments in section 6 with one of the monolingual sentiment Lexia. This experiment clarifies whether using SentiVAE helps to identify gender bias in the language models or not.

We agree that the validity of our results depends on the reliability of the sentiment lexica. Therefore, as mentioned above, we have extended Section 4.3 to have a more detailed explanation of the algorithm and our reasoning for using it in our analysis. The selected lexica we used are all publicly available and have been previously verified in a downstream evaluation on a sentiment classification task. Ultimately, we want to maximize the coverage of the words while retaining high quality of the lexicon used. However, due to disparities in their scales, we resort to the SentiVAE method to combine them. In the initial experiments, we have tested different combinations of lexica evaluating them on the listed sentiment analysis datasets. This analysis was a base to excluding certain lexica for Hindi, as mentioned in the paper. The lexicon evaluation results for Spanish are indeed better than the results of a state-of-the-art supervised algorithm. Hence, although they might seem relatively low, we assume that they deliver results of as high a quality as it is currently possible. Further, the previously curated lexica were created using human annotations and therefore, we do not see added value from using more human annotation on the data. SentiVAE is solely a transformation of the lexica from their initial scale to be represented as a vector with values for three sentiments, positive, negative and neutral. An ablation study that was suggested by the reviewer would not verify the lexicon since we do not have an explicit information of ground truth gender bias to compare it to report how much of it we detect using our lexicon. 

I still recommend the authors to provide a more in-depth analysis that would go beyond the experiments and the results of the previous studies. The paper studies an important question and the authors have collected valuable data to investigate the question, but the experiments and the results provide only a surface-level analysis of this problem.

We’re glad that the reviewer recognises the importance of our research topic. As indicated above, following their recommendation, we have extended the experimental section of the paper. We believe that the new results increase the value of our paper and provide valuable insights about the nature of bias encoded in language models across the analyzed languages.

The readability of some parts, especially the methodology and the experiments, should be improved as well.

We have rewritten the methodology section and added more detailed explanations of models used. We have added further experiments which we describe above. 

Writing comments:

In section 6.1., the authors could provide a description of the supersets used and the set of words each contains.

We have added a description of the supsersets together with words they include in the Appendix.

I find it difficult to understand the first paragraph of section 6.1.2. Could you provide some examples for each step mentioned in this section?

We have rewritten this part of the paper and added a step by step explanation, and examples. 

7. PLOS authors have the option to publish the peer review history of their article (what does this mean?). If published, this will include your full peer review and any attached files.

Do you want your identity to be public for this peer review? For information about this choice, including consent withdrawal, please see our Privacy Policy.

Reviewer #1: No

Reviewer #2: No

---

## [Decision Letter · Decision Letter 2]

2 Nov 2022

Quantifying Gender Bias Towards Politicians in Cross-Lingual Language Models

PONE-D-22-03986R2

Dear Dr. Stanczak,

We’re pleased to inform you that your manuscript has been judged scientifically suitable for publication and will be formally accepted for publication once it meets all outstanding technical requirements and the final comments raised by the reviewer.

Kind regards,

Yang Xu, PhD

Academic Editor

PLOS ONE

Additional Editor Comments (optional):

I have received final comments from Reviewer 2, who indicated that the authors have managed to address the main issues raised in the previous revision. I will accept the manuscript for publication if the authors can aim to address the lingering (more minor) issues raised by the reviewer in the final version, quoted as follows:

1) In Table 4, why is the word "great" in Tmasc_neg? Similarly, why "first" and "out" are in this column? There seem to be some noises in the proposed methodology, even in the top 10 adjectives. Some explanation or error analysis is needed here.

2) Where are the results on other languages and language models for the clustering experiment in Table 7? Currently, it is only done (or reported) for French and m-BERT.

3) The clustering and PCA experiments could both be discussed in more detail. There are no significant patterns discovered. How does this correspond to the results of the previous sections? (Other than the fact "that more gendered words are generated more frequently in association with their respective gender", which is expected regardless of the gender biases).

4) The quality of the images could be improved (especially Figure 3, and Figure 6).

Additionally, I'd encourage the authors to offer some discussion on the language-specific effects found in their study---why gender bias patterns turned out the way they did, e.g. prior to or within the conclusive paragraphs.

Wording suggestion in Abstract: "we use to effect a multi-lingual study of gender bias towards politicians ..." ==> "we use to enable a multi-lingual study of gender bias towards politicians ..."?, or perhaps "we use to facilitate a multi-lingual analysis of gender bias towards politicians ..."

Reviewers' comments:

Reviewer's Responses to Questions

**Comments to the Author**

1. If the authors have adequately addressed your comments raised in a previous round of review and you feel that this manuscript is now acceptable for publication, you may indicate that here to bypass the “Comments to the Author” section, enter your conflict of interest statement in the “Confidential to Editor” section, and submit your "Accept" recommendation.

Reviewer #2: (No Response)

2. Is the manuscript technically sound, and do the data support the conclusions?

Reviewer #2: Yes

3. Has the statistical analysis been performed appropriately and rigorously? 

Reviewer #2: Yes

4. Have the authors made all data underlying the findings in their manuscript fully available?

Reviewer #2: Yes

5. Is the manuscript presented in an intelligible fashion and written in standard English?

Reviewer #2: Yes

6. Review Comments to the Author

Reviewer #2: The authors have addressed my previous comments by adding new experiments and have improved the clarity of their method section.

I have some minor comments and questions for the authors that perhaps need to be addressed in the paper before submission.

1) In Table 4, why is the word "great" in Tmasc_neg? Similarly, why "first" and "out" are in this column? There seem to be some noises in the proposed methodology, even in the top 10 adjectives. Some explanation or error analysis is needed here.

2) Where are the results on other languages and language models for the clustering experiment in Table 7? Currently, it is only done (or reported) for French and m-BERT.

3) The clustering and PCA experiments could both be discussed in more detail. There are no significant patterns discovered. How does this correspond to the results of the previous sections? (Other than the fact "that more gendered words are generated more frequently in association with their respective gender", which is expected regardless of the gender biases).

4) The quality of the images could be improved (especially Figure 3, and Figure 6).

7. PLOS authors have the option to publish the peer review history of their article (what does this mean?). If published, this will include your full peer review and any attached files.

Reviewer #2: No

---

## [Editor Report · Acceptance letter]

8 Nov 2023

PONE-D-22-03986R2 

Quantifying Gender Bias Towards Politicians in Cross-Lingual Language Models 

Dear Dr. Stanczak:

I'm pleased to inform you that your manuscript has been deemed suitable for publication in PLOS ONE. Congratulations! Your manuscript is now with our production department. 

Kind regards, 

on behalf of

Dr. Yang Xu 

%CORR_ED_EDITOR_ROLE%

PLOS ONE